# A Comparative Analysis of Post-Retraction Changes in Gingival Height after Conventional and Surgical Gingival Displacement: Rotary Curettage, Diode and Er:YAG Laser Troughing

**DOI:** 10.3390/healthcare11162262

**Published:** 2023-08-11

**Authors:** Rada Kazakova, Angelina Vlahova, Georgi Tomov, Mariya Dimitrova, Stoyan Kazakov, Stefan Zlatev, Marta Forte, Giuseppe Barile, Massimo Corsalini, Saverio Capodiferro

**Affiliations:** 1Department of Prosthetic Dentistry, Faculty of Dental Medicine, Medical University–Plovdiv, 4000 Plovdiv, Bulgaria; angelina.vlahova@mu-plovdiv.bg (A.V.); stefan.zlatev@mu-plovdiv.bg (S.Z.); 2CAD/CAM Center of Dental Medicine, Research Institute, Medical University–Plovdiv, 4000 Plovdiv, Bulgaria; 3Department of Periodontology and Oral Mucosa Diseases, Medical University–Plovdiv, 4000 Plovdiv, Bulgaria; georgi.tomov@mu-plovdiv.bg; 4Laser Dental Center, Research Institute, Medical University–Plovdiv, 4000 Plovdiv, Bulgaria; 5Oral Surgeon, Private Dental Practice–Sofia, 1000 Sofia, Bulgaria; kazakovstoyan@gmail.com; 6Department of Interdisciplinary Medicine, ‘Aldo Moro’, University of Bari, 70100 Bari, Italy; fmarta@live.it (M.F.); massimo.corsalini@uniba.it (M.C.); saverio.capodiferro@uniba.it (S.C.)

**Keywords:** gingival displacement, gingival retraction, laser troughing, pre-impression troughing, laser retraction, laser displacement, diode laser, Er:YAG laser, rotary curettage, ceramic bur

## Abstract

The aim of the current article is to analyze and compare post-retraction gingival height changes resulting from six different types of gingival-displacement methods, encompassing both conventional and surgical approaches. The study involved a comparative analysis of 263 teeth (consisting of 128 front teeth, 69 premolars, and 66 molars) from 23 patients. For the investigation, three classic retraction methods were utilized, namely the single-cord technique, retraction paste Expasyl, and retraction paste Astringent. Additionally, three surgical techniques were employed, which included ceramic bur rotary curettage, Er:YAG laser troughing, and diode laser troughing. A randomized split-mouth design was implemented, and a significance level of 0.05 was used for the study. The recovery of the free gingival margin height was assessed on gypsum models that were scanned using an intraoral scanner during the first and second week after the retraction procedure. The results revealed that all retraction methods, except for ceramic bur rotary curettage, led to clinically insignificant levels of gingival recession. The article provides insights into the effectiveness and safety of various gingival-displacement techniques, highlighting that most methods tested in the study resulted in minimal or negligible gingival recession post-retraction.

## 1. Introduction

Gingival retraction (gingival displacement) is the process of a reversible deflection or removal of the inner surface of the gingival sulcus. The purpose is to let the impression material cover, on the intraoral scanner scan, the prepared and unprepared part of the tooth. According to the Glossary of Prosthodontic Terms, 2017, it is ‘the deflection of the marginal gingiva away from a tooth’ [1].

The primary objective is to achieve proper and reversible gingival displacement while effectively handling the gingival tissues. This aims to achieve accurate reproduction of the finish line and the untouched unprepared tooth structure in the apical direction. Additionally, it involves creating a dry environment for the impression material or intraoral scanner, all while maintaining the contour, position, and periodontal health of the gingiva.

Gingival displacement methods can be broadly categorized into two main groups:Conventional mechanical, chemical, and mechano-chemical methods: These approaches involve techniques that use physical instruments, chemical agents, or a combination of both to displace the gingival tissues for impression taking. Examples may include retraction cords, astringent pastes, and retraction pastes containing specific compounds like aluminum chloride and kaolin.Surgical (troughing) methods: This group includes surgical techniques such as troughing, where a dental professional creates a trough or groove in the gingival tissue to facilitate proper retraction for accurate impression making.

The ultimate goal of these methods is to enable precise and efficient impression procedures while safeguarding the gingival health and preserving the natural structure of the gingiva.

Comparing post-retraction gingival height changes after conventional versus surgical displacement holds significant clinical importance for dental practice and patient outcomes. Understanding the differences between these two methods helps dentists tailor treatment plans based on individual patient needs, ensuring the most suitable technique is chosen for each case. It also helps in preserving gingival health by minimizing the risk of significant gingival recession. By selecting methods that cause less discomfort, patient satisfaction and treatment compliance can be improved. Additionally, the comparison aids in optimizing time efficiency and ensuring faster results when needed. Dentists can also make safer choices by considering the potential complications associated with surgical methods. Furthermore, evaluating the costs of different approaches helps in determining the most cost-effective option without compromising treatment quality. This comparison provides evidence-based insights for better patient care, comfort, and long-term oral health.

The aim of the present article is to evaluate and compare post-retraction gingival height changes resulting from six different types of gingival-displacement methods. These methods fall into two categories: conventional (mechanical, chemical, and mechano-chemical) and surgical approaches. The main objective of the study is to assess the effectiveness and safety of each method from a clinical standpoint. By examining the range of applications and potential side effects of these techniques, the research seeks to provide valuable insights for dental professionals in choosing the most suitable and reliable gingival displacement method for their clinical practice.

## 2. Materials and Methods

### 2.1. Object of Observation

The changes in the gingival height after each one of the retraction methods.

### 2.2. Units of Observation

Retracted gingiva in 263 teeth of 23 patients.

### 2.3. Parameters of Observation

The factor variables are as follows:○The six tested retraction methods;○Sex;○Age;○Group of teeth (frontal, premolars and molars);○Time of gingival recovery (first week and second week).

The resultant variables are as follows:○Presence of gingival recession (GR);○Hyperplasia of the free gingival margin level.

### 2.4. Settings and Location Where the Data Were Collected (Venue of Observation)

○Department of Prosthetic Dentistry, Faculty of Dental Medicine, CAD/CAM Center of Dental Medicine, Research Institute, Medical University—Plovdiv (RIMUP).○Department of Periodontology and Oral Mucosa Diseases, Laser Dental Center, Faculty of Dental Medicine, Research Institute, Medical University—Plovdiv, Bulgaria (RIMUP).

### 2.5. Eligibility Criteria for Participants in the Study

○Patients should not have systemic diseases. They may lead to inflammation and/or bleeding, compromising the results [2,3].○They should not have taken medications in the last three months. Some medications may cause inflammation and/or bleeding, compromising the results [2].○The Löe and Silness gingival index should be 0. The index assesses the prevalence and severity of gingivitis in populations, groups, and individuals. A score from 0.1 to 1.0 signifies mild inflammation, 1.1 to 2.0 moderate inflammation, and 2.1 to 3.0 severe inflammation [2].○Patients subject to prosthetic restorations in more than one quadrant.○Criteria for exclusion from the study are as follows: Patients with psychological disorders. The presence of inaccurate obturations can interfere with the biological width.○Teeth with a periodontal probing depth (PPD) above 3 mm.

The study was approved with a statement by the Committee of Scientific Ethics at Medical University—Plovdiv, reference number P-7350/01.10.2015. Patients signed informed consent and received a document about the study and the possible risks.

### 2.6. Entry of Primary Data

The reported gingival height (in mm) with the Trios CAD/CAM intraoral scanner (3Shape, København, Denmark) was also registered by photo documentation, as the photographic material was analyzed according to the tested variables. The obtained mean values (from the triple measuring of the gingival height) for the first and second weeks and the factor variables were plotted in spreadsheets for further statistical analysis.

### 2.7. Trial Design

A comparative analysis of post-retraction changes in gingival height after six different retraction methods was performed. A randomized ‘split mouth design’ study was conducted. The dentition was divided into four quadrants, and the gingival retraction of the opposite quadrants was performed using one of the six analyzed methods: mechano-chemical (three classic approaches) or surgical (three approaches). The choice of which retraction method would be used for a given quadrant was random—using computer-generated randomization.

Twenty-three patients, subject to fixed restorations in the Departments of Prosthetic dentistry, Faculty of Dental medicine, Medical University—Plovdiv, Bulgaria were examined (263 teeth in total). The participants were healthy patients subject to prosthetic restorations in more than one quadrant. The subjects were instructed in advance to maintain good oral hygiene. The first visit included teeth preparation and impression taking. The finish line was subgingival—chamfer, positioned 0.5 mm apically from the gingival margin. Temporary composite crowns were made and cemented on the prepared teeth.

Pre-study gingival dimensions served as the starting point, providing a reference measurement before any retraction procedures were conducted. This baseline is vital for accurately assessing the changes caused by different gingival-displacement methods. Having pre-studied gingival dimensions allows for a direct comparison with post-retraction measurements, and enhances the scientific validity of the research.

The following gingival displacement methods were applied at random:Classic mechano-chemical:
○Retraction cords Elite cord (Zhermack, Badia Polesine, RO, Italy) in five different sizes (000, 00, 0, 1, 2) depending on the sulcus depth, impregnated with a 5% aluminium chloride solution.○Retraction paste Expasyl (ACTEON Pharma – Pierre Rolland, Mérignac, France).○Retraction paste Astringent (3M ESPE, Seefeld, Bayern, Germany).
Surgical:
○Ceramic bur Soft Tissue Trimmer NTI (Kerr, Orange, CA, USA).○Er:YAG laser with a wavelength of 2940 nm (Light Instruments, Yokne’am Illit, Israel).○Diode laser with a wavelength of 810 nm and power of 8W (FOX, A.R.C. Lasers GmbH, Nürnberg, Germany).


The time of the retraction for each technique was also standardized according to well-established protocols. For the classical methods, the cord and the pastes stayed for 10 min in the sulcus before the impression taking to ensure sufficient displacement. As far as the surgical methods were concerned, the impression was taken 10 min after the retraction to guarantee excellent haemostasis.

The distance from the free gingival margin to the centre of a sphere, made with a round bur (0.08 mm diameter) on the facial side of the tooth of the patient, was measured.

Three measurements were made on scanned plaster models cast from the impressions:○1st measurement—immediately after the retraction—the value was considered 0.○2nd measurement—one week after the procedure.○3rd measurement—two weeks after that.

In the virtual cut through the scanned tooth, the distance between the deepest point of the facial mark and the gingival edge was measured. Gingival recovery or occurrence of possible gingival recession was assessed in the two post-retraction stages of the measurement. The obtained mean values of the gingival height in every retraction method were compared.

### 2.8. Technique of Implementation of the Applied Gingival Displacement Methods

It is established that plaque accumulation can lead to gingivitis and periodontitis, which can compromise the results. The subjects were instructed in advance to maintain good oral hygiene by using contemporary mechanical and chemical methods of plaque control to prevent gingivitis [2]. The plaque index of the patients was 0, indicating no visible plaque [2].

1. Retraction cords Elite cord (Zhermack, Badia Polesine, RO, Italy) in five different sizes (000, 00, 0, 1, 2) depending on the sulcus depth were used for the single-cord technique. The cord was impregnated with a 5% aluminum chloride solution. A smooth packing instrument was carefully used to place the cord in the sulcus to ensure excellent visual access to the finish line. The packing of the cords did not cause hemorrhage or tearing of the marginal gingiva. Right before taking the impression, the cord was watered to prevent gingival tearing and removed, and the sulcus was washed and air-dried.

2. The second type of retraction was performed using Expasyl retraction paste (ACTEON Pharma—Pierre Rolland, Mérignac, France) with a special applicator gun and a cannula. The retraction paste contains 15% aluminum chloride and kaolin. It was placed in the sulcus around the circumference, filling it to the bottom. After 2 min of waiting for volume expansion, it was washed away thoroughly immediately before taking the impression.

3. The third type of retraction was using Astringent Paste (3M ESPE, Seefeld, Bayern, Germany) in a compule with a special applicator gun. The retraction paste contains 15% aluminum chloride. It was placed in the sulcus around the circumference so that it filled it to the bottom. After 2 min of waiting for volume expansion, it was washed away thoroughly immediately before taking the impression.

4. The fourth type of retraction was the ceramic bur NTI Soft Tissue Trimmer (Kerr, Orange, CA, USA) rotary curettage. Unlike the classic rotary curettage with a diamond bur, when the preparation is finished, and the sulcus is simultaneously de-epithelized, the ceramic bur cuts soft tissues only. It works with a turbine without water cooling. It slid through the sulcus around the circumference, removing the superficial epithelium. Coagulation, therefore, of the gingival tissues’ hemostasis was achieved due to the temperature rise. Enough epithelium was removed to leave space for the impression material and to visualize the finish line. Before taking the impression, the sulcus was dried, so the hydrophobic impression material could fill it up.

5. The fifth type of retraction was performed using the Er:YAG LiteTouch laser (Light Instruments, Yokne’am Illit, Israel) with a wavelength of 2940 nm. A thin chisel tip 1.3 × 17 mm was used in contact mode, with an incessant brushing movement and a 15° angle to the root surface. The settings were the following: Soft Tissue Mode, 50 mJ, 10 Hz (0.50 W altogether), with incessant water cooling. Enough epithelium was removed to leave space for the impression material and to visualize the finish line. Before taking the impression, the sulcus was dried, so the hydrophobic impression material could fill it up.

6. The sixth type of retraction used a diode laser (FOX, A.R.C. Lasers GmbH, Nürnberg, Germany) with a wavelength of 810 nm and a maximum power of 8 W. The tip of the 300 µm fiber was activated by holding it on a special black paper prior to starting the procedure. The retraction always started at lower power. The diode laser was set to a continuous mode with a power of 1.5 W and, if necessary, it could be increased to 2 W. The working mode was Gingivectomy. Tissue charring was a side effect of using the laser and could occur either because of operating at too high a power or because the tip moved too slowly.

The optical tip, held at an angle to the soft tissues and away from the prepared tooth, slid by the gingival sulcus to remove the sulcus epithelium. Constant and stable short brushing moves carefully removed the inner sulcus epithelium to provide a 360-degree trough. The tip was cleaned with wet gauze, and soaked in 3% hydrogen peroxide, to remove the debris and eliminate possible bacterial contamination. Enough epithelium was removed to leave space for the impression material and to visualize the finish line. Before taking the impression, the sulcus was dried, so the hydrophobic impression material could fill it up.

Following the laser safety protocols, the researchers and the patients were protected from laser irradiation by safety goggles for the specific laser wavelength.

### 2.9. Impression Materials

For the conducted study, a selection of impression materials was employed to fulfill the research objectives. Preliminary impressions were taken using alginate, specifically Tropicalgin (Zhermack, Badia Polesine, RO, Italy). Additionally, the study utilized addition silicones, also known as A-silicones, for its impression-taking purposes. The specific brand employed for the A-silicones was Variotime (Heraeus Kulzer, Hanau, Germany). After taking the preliminary alginate impressions, custom trays from light-cured resins were made to take more precise impressions and save impression material. The final impressions were taken with an A-silicone. The finish line was a subgingival chamfer, positioned 0.5 mm apically from the gingival margin. Temporary composite crowns were made and cemented on the abutment teeth.

The impressions were taken at the same visit as the retraction. An assessment of the gingival configuration was made with the help of the final impressions during the three beforementioned stages of the study.

The facial surface of all prepared teeth was marked at a 3 mm distance from the free gingival margin with a round bur with a fixed diameter (0.08 mm). The final impressions were taken with a custom tray and Variotime A-silicone.

In order to dose and equalize the pressure exerted during impression taking, at least three ‘occlusal stops’ were made on the custom trays. Two layers of pink wax were adapted to provide the necessary space for the impression material. The wax was cut at the areas of the occlusal stops, and a thin aluminum foil was adapted on top of it. The light-cured resin was then adjusted, the excesses were cut with a knife, and a handle was made using some of them. The impression tray was cured in a light-curing unit.

### 2.10. CAD/CAM System

Scanning the plaster models using the Trios intraoral scanner (3Shape CAD/CAM system) was performed to measure the free gingival margin level at every stage of the study (Figure 1). In order to obtain optimally accurate results, a threefold measurement of the free gingival margin was performed, and the values obtained were averaged. The measured length immediately after retraction was considered to be 0, and the next two measurements were at week one and week two, and tracked the changes in the free gingival margin height.

The scanning accuracy of the Trios 3 (3Shape, København, Denmark) intraoral scanner, according to American Dental Association (ADA) data, is 6.9 ± 0.9 µm [4].

A total of three impressions were taken: immediately after the retraction, at one week and then at two weeks. Plaster models were cast and then scanned with the Trios intraoral scanner. In order to achieve three-dimensional accuracy, with the help of the CAD/CAM system, a virtual cut through the middle of the measured tooth along its longitudinal axis was made (Figure 2).

The distance between the middle of the mark, made with the 0.08 mm bur on the prepared tooth, and the free gingival margin on the facial surface of the tooth was measured (Figure 3). Thus, possible changes in the gingival height level after the retraction were monitored.

### 2.11. Statistical Methods

The statistical methods were implemented with the statistical program SPSS version 19, and the results were considered statistically significant at α < 0,05. The statistical analyses applied were the following: descriptive analysis for describing the structures of the given variables; *X*^2^-anaysis (Chi-squared test) for establishing dependencies between qualitative variables; Mann–Whitney U-Test for comparison between two samples (classical retraction methods and surgical retraction methods) in which the distribution of quantitative variables is not normal; and graphical analysis for illustration of the results obtained.

## 3. Results

### 3.1. Sample Characteristics

The study compared the obtained mean gingival height levels after retraction with different methods of a total of 263 teeth (of 23 healthy subjects), 44.5% of the analyzed were women (117 teeth), and 55.5% were men (146 teeth).

The average age of the patients was 36.20 ± 6.039 years old (the youngest was 18 years old, and the oldest was 56 years old). The average age of the women was 33.41 ± 4.27 years old, and men was 38.44 ± 6.32 years old. The mean difference (5.03 years) was statistically significant (u = 7.369, *p* = 0.000). Please see Table 1 for reference.

The three classic mechano-chemical retraction methods were applied on 48.3% of teeth (127 teeth), and the three surgical ones on 51.7% of teeth (136 teeth).

Figure 4 (Appendix A, Table A1) shows the structural distribution of teeth retracted with a specific method. The mean number teeth retracted with each of the six methods was 42 to 46.

In Figure 5 (Appendix B, Table A2), the number and percentage of the examined teeth are presented, depending on whether they are frontal, premolars or molars; the frontal ones represent 48.7% of teeth (128 teeth), the premolars 26.2% (69 teeth), and the molars 25.1% (66 teeth). A similar number of frontal and distal teeth was sought to be achieved for the sake of the study analyses. Appendix C (Table A3) shows the structural distribution of the examined teeth in detail.

### 3.2. Experimental Results

The mean value of the measured gingival recession at the first measurement (the first week after the retraction) for the classic mechano-chemical methods was 0.011 ± 0.17 mm; the surgical methods presented with a more notable, but not clinically significant, recession of 0.23 ± 0.25 mm (Table 2). The intergroup difference between the compared levels of the free gingival margin was 0.24 mm, which was statistically significant u = 8.954, *p* = 0.00.

The measured gingival recession value in the second week after the retraction was 0.02 ± 0.13 mm for the classic mechano-chemical methods and 0.15 ± 0.21 mm for the surgical ones. The discovered significant difference was 0.14 mm, which was approximately twice less than the difference found between the two methods in the previous measurement.

Table 3 presents the mean values of gingival height in the first and second week of the two main methods used, depending on whether they were applied on front teeth, premolars and molars. The differences in gingival height between the two main compared retraction methods were found to be more significant in the premolars and molars than in the frontal teeth.

Table 4 presents the mean gingival level at the two measurements. The table demonstrates which methods which, in particular, resulted in gingival recession or an increase in the free gingival margin level. The data showed that the methods with the Astringent Paste (a retraction paste containing 15% aluminum chloride) and the Expasyl paste (a retraction paste containing kaolin and 15% aluminum chloride) resulted in a slight, clinically inexpressible post-retraction gingival hyperplasia.

The other remaining four retraction methods resulted in a different degree of recession, which was clinically significant only for the patients that underwent ceramic bur rotary curettage, respectively, 0.45 ± 0.27 mm for the first week and 0.32 ± 0.23 mm for the second one (Table 4). The lowest recession levels were recorded at the Er:YAG retraction (0.10 ± 0.13 mm for the first week and 0.06 ± 0.10 mm for the second one).

Because both statistically and clinically significant higher gingival recession levels were obtained after the ceramic bur retraction (surgical method) (Table 5), this method was excluded from the analysis, and the combined mean values of the gingival height for the other two surgical (laser) methods were compared. After the change in Table 4 (the rotary curettage data was excluded), it could be seen that the reported gingival recession levels in the first and second week for the surgical (laser) methods dropped almost twice compared to the data from Table 6. The differences in the gingival recession levels were also twice as low in this comparative approach.

Table 6 presents the specific gingival recession or hyperplasia values when grouping the teeth as frontal, premolars and molars, depending on the specific retraction methods used. It should be noted again that the highest clinically significant gingival recession value was reported in the ceramic bur rotary curettage group. The highest gingival recession level was reported for premolars 0.56 ± 0.32 mm for the first week and 0.41 ± 0.29 mm for the second one. The recession levels for this method were slightly lower for the molars—0.51 ± 0.21 mm and 0.35 ± 0.23 mm for the first and the second week, respectively.

The Er:YAG laser group demonstrated the lowest loss of gingival height levels in every single group of teeth (Table 6). The mean recession level in the diode laser retraction group was twice as low as in the mechano-chemical group (retraction cord) for the frontal teeth. The mean gingival height loss after diode laser retraction in the premolars and molars groups was slightly more than after the retraction cord. All the differences found in the gingival height measurement between the compared methods in the three groups of teeth were statistically significant.

## 4. Discussion

Numerous authors have focused on various retraction techniques employed in dentistry and the subsequent extent of undesirable gingival recession, as documented in references [5,6,7,8,9,10,11,12,13,14,15,16,17,18,19,20].

Clinical and technical characteristics of these retraction methods indicate their respective advantages and disadvantages in dental practice.

The commonly used double-cord technique for subgingival tissue displacement involves removing the upper cord prior and injecting the impression material or taking the optical impression to ensure a thorough reproduction of the finish line. However, this method is time-consuming and has certain drawbacks. These include the possibility of damaging the periodontal ligament if excessive force is applied during the process, the difficulty in removing the retraction cord without causing rupture or bleeding, and potential postoperative discomfort [21].

Based on the findings of our study, the groups that utilized two different retraction methods showed the least impact on the free gingival margin level. These methods were the following:Astringent Paste: This retraction paste contains 15% aluminum chloride.Expasyl: This retraction paste contains kaolin and 15% aluminum chloride.

Clinically insignificant gingival hyperplasia was observed in these groups. The measurements taken during the first and second weeks showed minimal changes in the gingival margin level. Specifically, for the first period, the changes were 0.12 ± 0.11 mm and 0.06 ± 0.06 mm, respectively. For the second period, the changes were 0.06 ± 0.05 mm and 0.01 ± 0.01 mm, respectively (as shown in Table 4). It is worth noting that this retraction technique resulted in extremely low trauma to the gingival tissues compared to the traditional retraction cords. Other researchers have also acknowledged and reported similar findings in their studies [12,20,22].

Additionally, our study demonstrated fast recovery within a week after the impression-taking process [12,23]. This means that the subjects involved in the study showed significant improvement and restoration of their gingival tissues within just one week after undergoing the retraction procedure. Our study findings support the observation that ceramic bur rotary curettage is the most traumatic retraction method for gingival tissue [18,24,25]. In our research, we observed the most significant and clinically relevant gingival recession when this technique was used. During the first week after the procedure, the gingival recession measured 0.45 ± 0.27 mm, and in the second week, it decreased slightly to 0.31 ± 0.23 mm (as shown in Table 4). These results indicate that the ceramic bur rotary curettage approach had a more pronounced and lasting impact on the gingival tissues compared to the other retraction methods evaluated in our study. Among the various tissue retraction lasers available, the most frequently employed ones are diode lasers, followed by erbium lasers. These laser-based approaches offer several significant advantages, including the following:Lack of subsequent gingival inflammation: The use of diode and erbium lasers results in reduced or negligible gingival inflammation following the procedure. This is beneficial as it helps in minimizing discomfort and promotes faster healing.Minimal to no pain during the procedure: Patients undergoing laser retraction experience minimal to no pain, which can be particularly advantageous as it often eliminates the need for local anesthesia.Reduced gingival recession: Laser troughing leads to less gingival recession compared to traditional methods. This means that the gingival margin remains more stable and less tissue is lost.Reduced tissue bleeding: Laser retraction techniques result in decreased tissue bleeding during the procedure, contributing to a smoother and more controlled clinical experience.

These advantages make diode and erbium lasers popular choices for tissue retraction in dental practice, as they offer improved patient comfort and better clinical outcomes.

Stuffken et al., in 2016, conducted a study to assess the gingival recession level using two different approaches: 1. Double-cord technique with 5% aluminum chloride (mechano-chemical retraction), and 2. Diode laser with an 810 nm wavelength (surgical troughing). The study involved six patients, each requiring two crowns on intact teeth [26]. One tooth was randomly retracted using the double-cord technique with 5% aluminum chloride, while the other tooth was retracted using the diode laser. After eight weeks of the final crown cementation, the authors reported that the average gingival recession for teeth retracted with the first technique was 0.26 mm, ranging from 0.00 mm to 0.72 mm. In the second group, the average gingival recession was 0.27 mm, varying from 0.001 mm to 0.68 mm. The authors conclude that the observed recession values in both tested methods were similar and considered not clinically significant.

It is important to highlight that our study revealed significantly lower gingival recession values for the retraction method with the single cord impregnated with 5% aluminum chloride compared to the findings of Stuffken M et al. [26]. In our study, the gingival recession during the first week was only 0.15 ± 0.19 mm, and in the second week, it further reduced to 0.12 ± 0.17 mm. Similarly, our study’s data for the diode laser method indicated even lower levels of gingival recession than reported by Stuffken M et al. [26], with values ranging from 0.001 mm to 0.68 mm. Specifically, for the first week after retraction, our diode laser values were 0.14 ± 0.17 mm, and for the second week, they were 0.10 ± 0.18 mm.

Furthermore, our study revealed the lowest gingival recession level in the Er:YAG troughing group, with values of 0.10 ± 0.13 mm for the first week and 0.6 ± 0.10 mm for the second week (as shown in Table 4). It is worth noting that our results did not support the established risk of undesired gingival recession reported by Wilder-Smith P in 1997 [27].

Moreover, Ruel J et al., in 1980 [28], reported a mean gingival recession value of 0.2 ± 0.1 mm one week after the single-cord retraction technique, which was entirely comparable to our data on the occurrence of gingival recession in the single-cord retraction group. Our study’s findings indicate lower levels of gingival recession for the single-cord, diode laser, and Er:YAG troughing methods, which suggests their potential advantages in preserving gingival health and minimizing undesirable recession compared to previous research. Azzi R et al., 1983 [28] conducted a study where they analyzed three gingival retraction methods: retraction cord, electrosurgery, and rotary curettage. They concluded that the rotary curettage method led to more pronounced gingival recession, while the electrosurgical method showed minimal recession, and the retraction cord method had no recession. Our study also confirmed clinically significant gingival recession in the rotary curettage group [29]. However, in contrast to our results, Moskow B, in 1964, did not find significant differences in gingival recession between the retraction-cord method and rotary curettage [30]. After excluding rotary curettage data from the surgical methods, our study’s analysis showed comparable mean gingival recession levels between the three classic methods and two laser methods for the first and second week after retraction (as shown in Table 5).

Qureshi SM et al. [31], in 2020, conducted an in vivo study comparing three retraction methods: retraction cord impregnated with 25% aluminum sulfate solution, Expasyl retraction paste, and Astringent Paste. Their findings indicated that Astringent Paste resulted in the highest retraction (0.50 mm), followed by the retraction cord (0.48 mm) and Expasyl (0.34 mm). However, Prasanna GS (2013) reported better performance of Expasyl compared to the classic retraction cord [32]. In another study by Shrivastava KJ et al., in 2015, comparing a mechanical (Magic Foam cord) and two chemical–mechanical methods (Expasyl paste and retraction cord impregnated with 15% aluminum chloride), all three methods were clinically effective, providing 0.2 mm displacement for elastomeric impression materials [33]. Thimmappa M et al., in 2018, found that the merocel strip provided maximum vertical and horizontal retraction compared to Ultrapak Retraction Cord and Magic Foam, respectively [34]. Other authors also conducted comparative evaluations of non-invasive cordless retraction methods, which showed sufficient displacement for impression-taking [35,36,37].

Nasim H et al., 2023 [38] performed a randomized controlled clinical trial comparing the conventional retraction cord method with a novel method using polytetrafluoroethylene (PTFE). They measured mean horizontal gingival displacement using a stereomicroscope and assessed post-displacement bleeding and ease of application. While gingival displacement was similar between the two groups, the PTFE group experienced more post-retraction bleeding and patient discomfort. Further research is needed to improve the biological reaction to PTFE cord.

Gajbhiye V et al., in 2019, conducted a comparison of gingival displacement using a retraction cord with two new polyvinyl siloxane (PVS) impression materials—Aquasil and NoCord VPS one-step self-retracting impression system. The retraction cord provided the maximum amount of retraction, followed by NoCord and Aquasil, respectively. All three methods resulted in a gingival displacement greater than 0.2 mm, sufficient for precise impression taking [39]. However, Mehta S, in 2019, compared gingival retraction with a copper-wire-reinforced cord, polyvinyl siloxane foam, and vinylpolysiloxane paste, and found that the new systems were not as effective as the conventional ones [40]. Similarly, Kesari ZI (2019) obtained comparable results, stating that the best outcomes were achieved using a retraction cord with Racegel containing 25% aluminum chloride. Nonetheless, there was no significant difference between the tested methods—retraction cord with Racegel, Viscostat Clear containing 25% aluminum chloride, Vasozine eyedrops containing 0.05% tetrahydrozoline, and Racegel without a cord [41]. Pressure also plays a role in the quality of impressions [42,43].

Schmitz JH et al., in 2020, demonstrated the efficacy of another gingival-displacement method—the interim restoration technique [44]. Tao et al. (2018) compared pre-saturated cord and lasers (diode, Nd:YAG, and Er:YAG)—commonly used gingival-troughing techniques. Their study involved 50 patients and 108 anterior teeth. The gingival retraction groups were pre-saturated cord, diode laser, Nd:YAG laser, and Er:YAG laser. Gingival width and gingival recession were measured at three separate intervals: immediately following displacement, one week later, and four weeks later. The pre-saturated cord resulted in narrower gingival sulci and substantially higher gingival recession compared to lasers (*p* < 0.05). Among the lasers, the Er:YAG laser showed the fastest and least uneventful wound healing when compared to diode and Nd:YAG lasers [45].

In their 2018 study, Goutham GB et al. [46] compared the clinical efficacy of three different gingival retraction systems on gingival sulcus width. The study involved 45 participants, with each participant having one maxillary central incisor treated. The three groups were as follows: retraction cord impregnated with aluminum chloride (Group I), Magic Foam (Group II), and diode laser (Group III). Each method was randomly applied to 15 patients. The results showed that Group III, which used the diode laser, resulted in the most displacement compared to the other two groups. The authors concluded that laser troughing was the most efficient retraction technique, but the choice of method should be based on the clinical situation and the dentist’s preference. According to Thomas MS et al. in 2011, clinicians should adapt their armamentarium and gingival-displacement techniques to meet specific requirements and achieve predictable results [47]. Melilli D et al., in 2018, conducted a comparison between two gingival retraction methods—retraction cord and diode laser [48]. They measured the clinical crown height of 74 abutment teeth, dividing them randomly into two groups for each method. Measurements were taken at multiple time points, including after tooth preparation, 15 days post-preparation, before exposing the finish line, 10 min after that, and 15 days after taking the final impression. The study evaluated the ease of the technique, patient comfort using the Visual Analogue Scale (VAS), time needed for the procedure, and bleeding. The authors concluded that there was no significant difference between the two methods in terms of clinical crown height differences, indicating similar levels of retraction and restoration to the initial situation. However, laser troughing was found to be quicker, easier for the clinician, and more comfortable for the patient.

Kurtzman GM et al., 2017 presented a clinical case report demonstrating the advantages of laser troughing in improving scanning and impression taking [49]. The therapeutic benefits of diode lasers in enhancing impressions for restorative procedures have been proven, as they are safe and efficient in promoting tissue repair. The increasing interest in soft tissue lasers among clinicians, as highlighted by Lee EA in 2006, is partly due to their potential utility in gingival pre-prosthetic procedures [50]. Soft tissue lasers, including diode lasers, show promise in controlling moisture and facilitating hemostasis, making them suitable for esthetic crown lengthening and gingival troughing. As dentists gain more experience with these technologies, their applications are expected to expand. Lee EA’s study, which used an 810 nm diode laser for perio-restorative treatments in the anterior maxilla, further supports this trend [50].

In the past, electrosurgery has been utilized as a retraction method to open the sulcular tissue around tooth preparations, serving as an alternative to retraction cords and pastes. However, due to its high voltage and deeper cell action, electrosurgery has been associated with soft tissue consequences, as documented in the literature [51]. A biometric and histometric analysis using four Rhesus monkeys was conducted to study electrosurgical gingival troughing with fully rectified current. The procedure led to a statistically significant recession of the free gingival margin and a loss of connective tissue attachment due to apical migration of the junctional epithelium. Burn traces from electrode interactions were observed on cemental surfaces and dentin near the cementoenamel junction. The experimental teeth exhibited apical migration of the junctional epithelium, likely due to the coverage of cemental burn marks by epithelium. Additionally, some experimental sites showed az minor loss of crestal alveolar bone, and in one case, a bone sequestrum developed. Interestingly, electrosurgical contact with the cemental surface triggered the formation of secondary dentin. In contrast, the diode laser’s use, which operates at a low power level, does not raise concerns about tissue shrinkage [49].

Ünalan Değirmenci B et al., in 2021, conducted a study to compare the clinical effects of three retraction systems on the gingival tissue [52]. The three systems evaluated were: an Ultrapak Retraction Cord, a cordless system containing 15% aluminum chloride hexahydrate—Traxodent—and an Er,Cr:YSGG laser device—WaterLase^®^ iPlus. Digital scanning was performed, and follow-up appointments were scheduled at various time points up to one year. Six periodontal scores were assessed, including probing depth (PD), plaque index, gingival index (GI), mobility, sensitivity, and bleeding on probing (BOP). The results showed that the retraction cord and the cordless system had similar effects on the periodontium in terms of GI and PD indices. GI tended to decrease from the first day up to the first month and then increase from the third month onwards, while the PD indices increased in both groups. Er,Cr:YSGG laser troughing was found to be less harmful to the periodontal tissues, exhibiting lower PD, GI, and BOP indices over the course of one year. Within the limitations of our study, we observed that frontal teeth exhibited two to three times less gingival recession after troughing with the two lasers compared to the single-cord technique, making laser retraction more suitable for these teeth (Table 6). The mean recession values in premolars and molars for the three compared approaches (retraction cord, erbium laser, and diode laser) were similar, although the GR in molars after diode laser troughing was slightly more pronounced than after the single-cord method (Table 6).

Goutham GB et al. reported no statistical difference in gender distribution and pre-displacement gingival width among the three retraction groups they studied and did not report any correlations with age [46]. As such, additional studies are needed to better understand the relationship between age, gender, and gingival-recession levels after retraction.

## 5. Conclusions

In conclusion, the comprehensive investigation conducted in our study sheds valuable light on the various gingival retraction methods and their impact on post-retraction gingival recession levels. The results indicate that the majority of the tested retraction techniques yielded clinically insignificant gingival recession, underscoring their effectiveness in minimizing undesirable periodontal changes.

Among the evaluated methods, the Er:YAG laser-troughing technique emerged as the most promising, exhibiting the lowest post-retraction gingival recession. This finding is of significant importance for dental practitioners, as it highlights the potential benefits of incorporating dental lasers, particularly the erbium laser, into their clinical practice to achieve more favorable gingival outcomes. On the other hand, the ceramic bur rotary curettage approach proved to be the most traumatic for the gingival tissues, resulting in clinically expressible and lasting gingival recession. As such, it is crucial for dentists to exercise caution and refrain from using this technique, especially for molars and premolars. It is important to emphasize the need for tailored treatment plans that consider potential implications on periodontal health.

While our study provides valuable insights into the realm of gingival retraction, it is essential to acknowledge its limitations. Further investigation and larger-scale studies are warranted to corroborate and expand upon our findings, allowing for a more comprehensive understanding of the complex dynamics between retraction methods and gingival recession.

In light of the study’s results, dental professionals are encouraged to consider the benefits of utilizing dental lasers, such as the Er:YAG and diode lasers, to achieve optimal gingival retraction and preservation of periodontal health. By leveraging the advantages of these modern techniques, clinicians can enhance patient comfort, minimize post-retraction complications, and deliver more precise and patient-centric restorative procedures.

Overall, our research provides a foundation for future studies in this field, with the ultimate goal of advancing dental practices and ensuring the best possible outcomes for patients undergoing gingival retraction procedures. By continuously refining and optimizing retraction methods, dental practitioners can contribute to the overall success and long-term satisfaction of their patients’ dental treatments.

## Figures and Tables

**Figure 1 healthcare-11-02262-f001:**
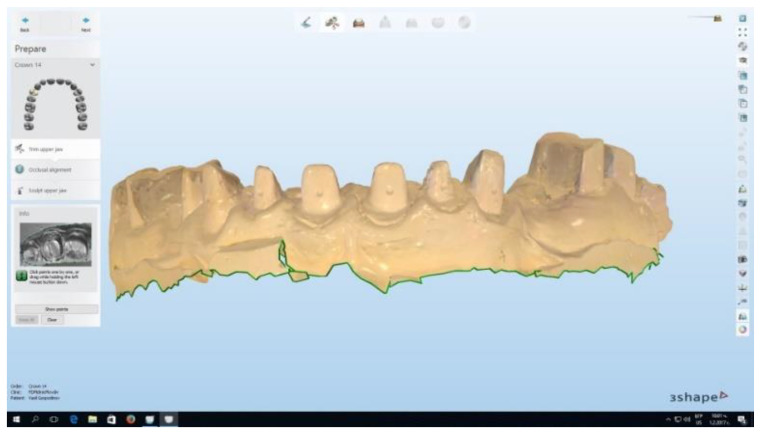
A scanned plaster model using the Trios (3Shape, København, Denmark) intraoral scanner.

**Figure 2 healthcare-11-02262-f002:**
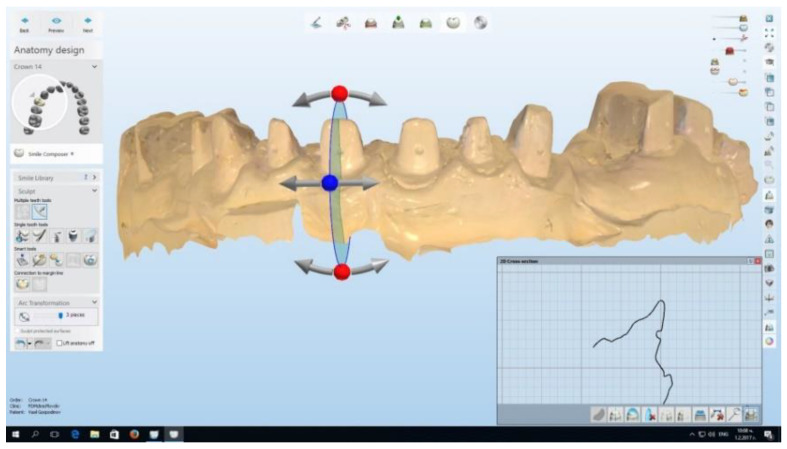
Virtual cut through the middle of the measured tooth. With the help of the vectors, the longitudinal axis of the tooth was accurately measured.

**Figure 3 healthcare-11-02262-f003:**
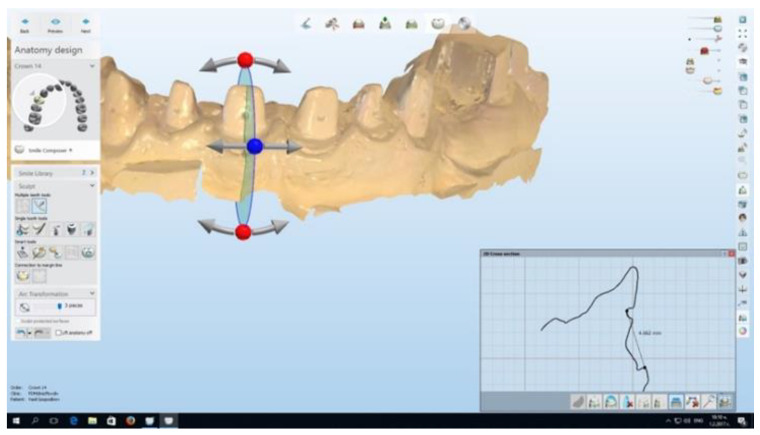
Technique of gingival height measurement.

**Figure 4 healthcare-11-02262-f004:**
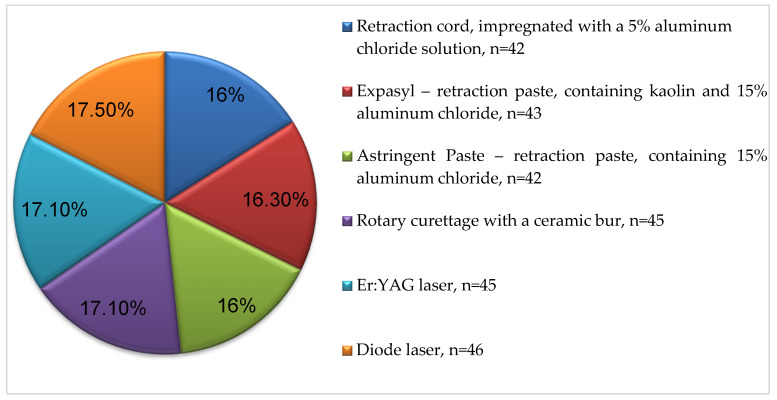
Structural distribution of teeth based on the applied retraction methods.

**Figure 5 healthcare-11-02262-f005:**
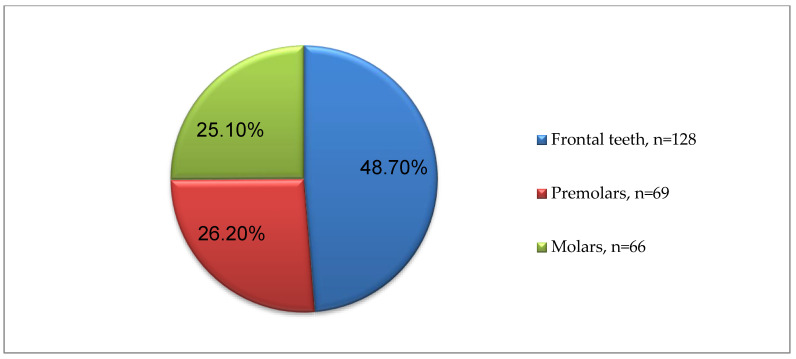
Grouping of the examined teeth.

**Table 1 healthcare-11-02262-t001:** Average (mean) age of the studied men and women.

	Sex	Number	Mean	Standard Deviation	Standard Error	U	*p*	Mean Difference
Age	Men	146	38.44	6.323	0.523	7.369	0.000	5.028
	Women	117	33.41	4.269	0.395			

**Table 2 healthcare-11-02262-t002:** Measured mean values of gingival recession depending on the methods applied.

	Methods	Number	Mean (mm)	Standard Deviation	u	*p*	Mean Difference
Values first week after the retraction	Classic mechano-chemical methods	127	0.011	0.170	8.954	0.000	0.239
	Surgical methods	136	0.228	0.251			
Values second week after the retraction	Classic mechano-chemical methods	127	0.017	0.127	6.443	0.000	0.137
	Surgical methods	136	0.155	0.206			

**Table 3 healthcare-11-02262-t003:** Measured mean total gingival recession values, depending on the main methods applied, when grouping the teeth.

		Methods	Number	Mean (mm)	Standard Deviation	u	*p*	Mean Difference
Frontal teeth	Values first week after the retraction	Classic mechano-chemical methods	63	−0.002	0.204	4.755	0.000	0.174
		Surgical methods	65	−0.176	0.209			
	Values second week after the retraction	Classic mechano-chemical methods	63	−0.031	0.162	3.022	0.003	0.091
		Surgical methods	65	−0.122	0.179			
Premolars	Values first week after the retraction	Classic mechano-chemical methods	33	0.027	0.154	5.148	0.000	0.304
		Surgical methods	36	−0.277	0.305			
	Values second week after the retraction	Classic mechano-chemical methods	33	−0.004	0.092	4.082	0.000	0.188
		Surgical methods	36	−0.192	0.249			
Molars	Values first week after the retraction	Classic mechano-chemical methods	31	0.018	0.097	6.149	0.000	0.292
		Surgical methods	35	−0.274	0.248			
	Values second week after the retraction	Classic mechano-chemical methods	31	−0.004	0.062	4.570	0.000	0.173
		Surgical methods	35	−0.177	0.201			

**Table 4 healthcare-11-02262-t004:** Measured mean gingival height values depending on every method applied.

Methods Applied		Values First Week after the Retraction	Values Second Week after the Retraction
Retraction cord, impregnated with a 5% aluminum chloride solution	Mean (mm)	−0.148	−0.124
	Number	42	42
	Standard deviation	0.185	0.167
Expasyl—retraction paste, containing kaolin and 15% aluminum chloride	Mean (mm)	0.116	0.057
	Number	43	43
	Standard deviation	0.110	0.056
Astringent—retraction paste, containing 15% aluminum chloride	Mean (mm)	0.062	0.013
	Number	42	42
	Standard deviation	0.047	0.012
Ceramic bur rotary curettage	Mean (mm)	−0.449	−0.307
	Number	45	45
	Standard deviation	0.269	0.227
Er:YAG laser	Mean (mm)	−0.100	−0.060
	Number	45	45
	Standard deviation	0.126	0.096
Diode laser	Mean (mm)	−0.137	−0.099
	Number	46	46
	Standard deviation	0.170	0.180
Total	Mean (mm)	−0.113	−0.088
	Number	263	263
	Standard deviation	0.246	0.185
	F	62.025	0.000
	*p*	33.923	0.000

**Table 5 healthcare-11-02262-t005:** Mean gingival height values measured depending on the methods applied (ceramic bur rotary curettage excluded).

	Two Main Methods (Ceramic Bur Rotary Curettage Excluded)	Number	Mean (mm)	Standard Deviation	u	*p*	Mean Difference (mm)
Values first week after the retraction	Classic mechano-chemical methods	127	0.011	0.170	5.795	0.000	0.130
	Surgical methods	91	−0.118	0.150			
Values second week after the retraction	Classic mechano-chemical methods	127	−0.017	0.127	3.341	0.001	0.062
	Surgical methods	91	−0.080	0.145			

**Table 6 healthcare-11-02262-t006:** Mean gingival recession or hyperplasia values, when grouping the teeth as frontal, premolars and molars, and depending on the specific retraction methods used.

			Number	Mean (mm)	Standard Deviation	F	*p*
Frontal teeth	Values first week after the retraction	Retraction cord, impregnated with a 5% aluminum chloride solution	21	−0.193	0.237		
		Expasyl—retraction paste, containing kaolin and 15% aluminum chloride	20	0.109	0.115		
		Astringent Paste—retraction paste, containing 15% aluminum chloride	22	0.080	0.048	25.689	0.000
		Ceramic bur rotary curettage	22	−0.361	0.254		
		Er:YAG laser	22	−0.075	0.072		
		Diode laser	21	−0.087	0.105		
		Total	128	−0.090	0.224		
	Values second week after the retraction	Retraction cord, impregnated with a 5% aluminum chloride solution	21	−0.168	0.215		
		Expasyl—retraction paste, containing kaolin and 15% aluminum chloride	20	0.063	0.069		
		Astringent Paste—retraction paste, containing 15% aluminum chloride	22	0.015	0.011	12.099	0.000
		Ceramic bur rotary curettage	22	−0.235	0.170		
		Er:YAG laser	22	−0.057	0.101		
		Diode laser	21	−0.072	0.201		
		Total	128	−0.077	0.176		
Premolars	Values first week after the retraction	Retraction cord, impregnated with a 5% aluminum chloride solution	11	−0.123	0.130		
		Expasyl—retraction paste, containing kaolin and 15% aluminum chloride	12	0.138	0.127		
		Astringent Paste—retraction paste, containing 15% aluminum chloride	10	0.060	0.041	17.774	0.000
		Ceramic bur rotary curettage	11	−0.561	0.317		
		Er:YAG laser	12	−0.132	0.207		
		Diode laser	13	−0.170	0.202		
		Total	69	−0.131	0.287		
	Values second week after the retraction	Retraction cord, impregnated with a 5% aluminum chloride solution	11	−0.090	0.106		
		Expasyl—retraction paste, containing kaolin and 15% aluminum chloride	12	0.058	0.052		
		Astringent Paste—retraction paste, containing 15% aluminum chloride	10	0.015	0.014	12.495	0.000
		Ceramic bur rotary curettage	11	−0.410	0.289		
		Er:YAG laser	12	−0.064	0.124		
		Diode laser	13	−0.125	0.179		
		Total	69	−0.102	0.212		
Molars	Values first week after the retraction	Retraction cord, impregnated with a 5% aluminum chloride solution	10	−0.081	0.055		
		Expasyl—retraction paste, containing kaolin and 15% aluminum chloride	11	0.103	0.084		
		Astringent Paste—retraction paste, containing 15% aluminum chloride	10	0.024	0.027	26.708	0.000
		Ceramic bur rotary curettage	12	−0.507	0.208		
		Er:YAG laser	11	−0.114	0.097		
		Diode laser	12	−0.187	0.213		
		Total	66	−0.137	0.241		
	Values second week after the retraction	Retraction cord, impregnated with a 5% aluminum chloride solution	10	−0.069	0.056		
		Expasyl—retraction paste, containing kaolin and 15% aluminum chloride	11	0.047	0.039		
		Astringent Paste—retraction paste, containing 15% aluminum chloride	10	0.005	0.013	14.546	0.000
		Ceramic bur rotary curettage	12	−0.343	0.230		
		Er:YAG laser	11	−0.057	0.048		
		Diode laser	12	−0.119	0.151		
		Total	66	−0.095	0.175		

## Data Availability

Not applicable.

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
