# Peer review of "A Comparative Analysis of Post-Retraction Changes in Gingival Height after Conventional and Surgical Gingival Displacement: Rotary Curettage, Diode and Er:YAG Laser Troughing"

_healthcare, 2023, doi:10.3390/healthcare11162262_

Round 1
Reviewer 1 Report
The submitted paper was well-written with detailed information on each retraction technique. There are minor points that improve the quality of the paper:
1. is this paper an RCT study? (is it registered in any clinical trial registration site?)
2. please mention the tip diameter for both diode and Er: YAG lasers.
3. how was the sample size calculated?
4. write the randomization process!
5. write about the safety protocols during laser irradiation (goggles..)
The submitted paper compared different retraction techniques that is well- designed and written in detail. I would recommend acceptance for publication after minor revision.
Author Response
Dear Reviewer,
Thank you for your valuable comments. We addressed them all.
- Yes, it is an RCT study – a randomized ‘split mouth design’ study was conducted. The dentition was divided into four quadrants, and the gingival retraction of the opposite quadrants was performed using one of the six analyzed methods: mechano-chemical (three classic approaches) or surgical (three approaches). The choice of which retraction method would be used for a given quadrant was random. It is not registered in a clinical registration site for now.
- Diode laser – 300 µm, Er:YAG laser – chisel tip, 1.3 x 17 mm [diameter x length]
- As a rule of thumb – the sample size is 10-20% of the full-scale sample size, or at least 30-50 objects (Pilot study, minimum 30 we calculated it using a sample size calculator – confidence level 95%, margin of error – 5%)
- The groups of the different retraction methods were randomly selected by applying statistical simple randomization – by randomly allocating the experimental units across the treatment groups. The randomization process was performed by a computer.
- Following the laser safety protocols, the researchers and the patients were protected from laser irradiation by safety goggles for the specific laser wavelength.
It took us a lot of time to finish our research, and we really put a lot of effort into this research. We think that the results are interesting and practical, that is why we want to share them with our colleagues. We truly hope that our article is accepted and will appreciate your help with that.
Best regards,
The Authors
Reviewer 2 Report
Authors fail to highlight the clinical significance of why addressing the comparison of post-retraction gingival height changes after conventional versus surgical displacement may be important. It is unclear, if the authors understand or are even able to clinically translate their own findings in conclusion, given the massive heterogenic changes occurring in the gingival displacement and recession process. The methods lack sufficient power owing to the many variables chosen. A power analysis a-priori would have been helpful. This needs to be re-executed in a larger cohort with a definitive power analysis. There is no mention about the clinician calibration which would greatly impact their study design. Another major flaw is the authors failed to highlight the importance of pre-study gingival dimensions to serve as baseline.
It is highly recommended to correct the typos throughout the manuscript and have the paper rewritten with the help of a native english speaker.
Author Response
Dear Reviewer,
Thank you for your valuable comments. We addressed them all. We do appreciate all your effort and the time you have spent reading the article.
We highlighted the clinical significance of the comparison. Comparing post-retraction gingival height changes after conventional versus surgical displacement holds significant clinical importance for dental practice and patient outcomes: ‘Understanding the differences between these two methods helps dentists tailor treatment plans based on individual patient needs, ensuring the most suitable technique is chosen for each case. It also helps in preserving gingival health by minimizing the risk of significant gingival recession. By selecting methods that cause less discomfort, patient satisfaction and treatment compliance can be improved. Additionally, the comparison aids in optimizing time efficiency, ensuring faster results when needed. Dentists can also make safer choices by considering the potential complications associated with surgical methods. Furthermore, evaluating the costs of different approaches helps in determining the most cost-effective option without compromising treatment quality. This comparison provides evidence-based insights for better patient care, comfort, and long-term oral health.’
We edited the conclusion section.
The calibration was performed similarly to other studies, cited in the bibliography, and is as precise as possible because of the digital measuring tools that were used. This is a pilot study and compared to other pilot studies, it has sufficient number of units for each of the treatment groups. We mentioned the limitations of our study.
Due to the digital measuring, we can evaluate pre-and post-study gingival dimensions. In the articles in the available literature, all the measurements are performed analogically. We consider digital measurements are a plus.
We emphasized the importance of the baseline gingival dimensions.
The typos and other mistakes were corrected, the paper was almost entirely rewritten.
We understand your point about too many groups, and we will approach our future research in a different way. We used that many groups because that were the methods that we used, and we thought the conclusions would be interesting. It took us a lot of time to finish our research and we really put a lot of effort into this research. We think that the results are interesting and practical, that is why we want to share them with our colleagues. We truly hope that our article is accepted and will appreciate your help with that.
Best regards,
The Authors
Reviewer 3 Report
The manuscript requires major revision that encompasses the abstract, methodology, and discussion. The study design should be clearly elaborated including the statistical analysis. The table format should be amended following the standard tables for the specific statistical tests when presenting the result. Typo errors are highlighted and further comments are noted in the reviewed manuscript.

Suggest for proof editing to improve the manuscript as a whole.
Author Response
Dear Reviewer,
Thank you so much for your guidance. It is the first time that we have seen such a detailed review. We will use your help for our future research. We do appreciate all your effort and the time you have spent reading the article. We addressed everything you pointed out.
We edited everything you highlighted in the pdf document. Please see the attached file with our replies.
We edited the abstract, methodology, study design and discussion as you suggested.
The table formats and everything else that needed to be put in the MDPI format, was also done.
Typo errors were corrected. We answered the comments in our pdf documents – please see the attached file.
We did the proof editing you suggested. Actually, the article is almost entirely rewritten.
We understand your point about too many groups, and we will approach our future research in a different way. We used that many groups because that were the methods that we used, and we thought the conclusions would be interesting. It took us a lot of time to finish our research, and we really put a lot of effort into this research. We think that the results are interesting and practical, that is why we want to share them with our colleagues. We truly hope that our article is accepted and will appreciate your help with that.
Best regards,
The authors

Round 2
Reviewer 2 Report
Thank you for the re-submission. The manuscript has been added with relevant changes/suggestions made by the authors
Author Response
Dear Reviewer,
Thank you for your valuable comments, your help with the article and for giving us the opportunity to share our results with our colleagues. We appreciate it.
Best regards,
The Authors
Reviewer 3 Report
Minor amendments are recommended based on the revised manuscript by the authors. The comments are stated in the attached file to further improve the manuscript.

Author Response
Dear Reviewer,
We thank you very much for your help and guidance. We edited everything you pointed out and highlighted. Please see the attached file. Literature sources about rotary curettage are scarce, it it not even described in some of the books, but we found it very important for our study, because this was one of our methods of retraction. Actually, ceramic bur rotary curettage is something new and unexplored. We thank you very much for allowing us to share our results with our colleagues.
Best regards,
The Authors
Best regards,
The Authors
